# Fine Particulate Matter-Induced Oxidative Stress Mediated by UVA-Visible Light Leads to Keratinocyte Damage

**DOI:** 10.3390/ijms221910645

**Published:** 2021-09-30

**Authors:** Krystian Mokrzyński, Olga Krzysztyńska-Kuleta, Marcin Zawrotniak, Michał Sarna, Tadeusz Sarna

**Affiliations:** 1Department of Biophysics, Faculty of Biochemistry, Biophysics and Biotechnology, Jagiellonian University, 30-387 Cracow, Poland; mokrzynskikrystian@gmail.com (K.M.); olga.krzysztynska@doctoral.uj.edu.pl (O.K.-K.); michal.sarna@uj.edu.pl (M.S.); 2Department of Comparative Biochemistry and Bioanalytics, Faculty of Biochemistry, Biophysics and Biotechnology, Jagiellonian University, 30-387 Cracow, Poland; marcin.zawrotniak@uj.edu.pl

**Keywords:** particulate matter, PM_2.5_, phototoxicity, oxidative stress, free radicals, singlet oxygen, skin, keratinocytes, skin aging, lipid peroxidation

## Abstract

The human skin is exposed to various environmental factors including solar radiation and ambient air pollutants. Although, due to its physical and biological properties, the skin efficiently protects the body against the harm of environmental factors, their excessive levels and possible synergistic action may lead to harmful effects. Among particulate matter present in ambient air pollutants, PM_2.5_ is of particular importance for it can penetrate both disrupted and intact skin, causing adverse effects to skin tissue. Although certain components of PM_2.5_ can exhibit photochemical activity, only a limited amount of data regarding the interaction of PM_2.5_ with light and its effect on skin tissue are available. This study focused on light-induced toxicity in cultured human keratinocytes, which was mediated by PM_2.5_ obtained in different seasons. Dynamic Light Scattering (DLS) and Atomic Force Microscopy (AFM) were employed to determine sizes of the particles. The ability of PM_2.5_ to photogenerate free radicals and singlet oxygen was studied using EPR spin-trapping and time-resolved singlet oxygen phosphorescence, respectively. Solar simulator with selected filters was used as light source for cell treatment to model environmental lightning conditions. Cytotoxicity of photoexcited PM_2.5_ was analyzed using MTT assay, PI staining and flow cytometry, and the apoptotic pathway was further examined using Caspase-3/7 assay and RT-PCR. Iodometric assay and JC-10 assay were used to investigate damage to cell lipids and mitochondria. Light-excited PM_2.5_ were found to generate free radicals and singlet oxygen in season-dependent manner. HaCaT cells containing PM_2.5_ and irradiated with UV-Vis exhibited oxidative stress features–increased peroxidation of intracellular lipids, decrease of mitochondrial membrane potential, enhanced expression of oxidative stress related genes and apoptotic cell death. The data indicate that sunlight can significantly increase PM_2.5_-mediated toxicity in skin cells.

## 1. Introduction

Skin is a natural barrier that contributes to the maintenance of the body’s homeostasis by protecting internal organs against harmful effects of various physical, chemical, and biological factors [1]. One of the physicochemical factors present in the surrounding environment that can disrupt skin homeostasis is smog [2]. Smog being a type of intense air pollution affects a significant part of the world’s population, especially those living in urban areas [3]. The main ingredient of smog is particulate matter (PM), which can be divided into three main categories: PM_10_, PM_2.5_, and PM_1_, representing particles of an aerodynamic diameter smaller than 10, 2.5, and 1 µm, respectively. Ambient particulate matter consists mostly of transition metal compounds (e.g., Fe(II), Cu (II)), adsorbed small reactive molecules, (e.g., environmentally persistent free radicals (EPFRs)), organic compounds (e.g., polycyclic aromatic hydrocarbons (PAHs)), minerals and soot [4,5].

Different compounds found in PM can exhibit photochemical activity and act as catalysts of ROS generation [6,7]. In the presence of light and hydrogen peroxide, redox-active metal ions such as iron and copper can generate hydroxyl radicals and possibly other reactive oxygen species (ROS) [6]. Moreover, certain semiconductors such as titanium dioxide (TiO_2_) and zinc oxide (ZnO) irradiated with visible or near-UV light can produce oxygen radicals and singlet oxygen [6,7,8]. Organic compounds including dyes, porphyrins, and aromatic hydrocarbons (e.g., benzo[a]pyrene) present in airborne pollution [9,10,11,12,13] can exhibit substantial photosensitizing ability to generate singlet oxygen.

The skin contains a number of chromophores including melanin pigments and carotenoids that scatter and absorb the incident light in a wavelength-dependent manner, leading to a reduction in the light energy density with the increasing skin depth [14]. Even though UVB radiation is mostly blocked by the stratum corneum, UVA radiation can penetrate the skin epidermis, and the penetration of blue light and green light in the skin can reach 1.5 mm and 3 mm, respectively, as demonstrated using Monte Carlo simulations [14]. Therefore, the modulatory effects of light should be taken into consideration when analyzing the toxicity of particulate matter in light-exposed tissues.

It has been reported that ambient particulate matter can not only penetrate through barrier-disrupted skin [15] leading to a ROS-dependent inflammatory response, but it also can induce skin barrier dysfunction [16,17] by down-regulating filaggrin via cyclooxygenase 2 (COX2) expression and prostaglandin E2 (PGE2) production [18]. Interestingly, recent in vivo studies in human subjects have shown that several pollutants can be taken up trans-dermally from air [19,20]. The solubility of certain compounds of ambient particles is a relevant factor influencing their toxicity and reactivity. Soluble compounds of PMs, such as nitrates or sulphates, can easily enter the cells causing adverse health effects [21,22], while insoluble compounds may induce ROS production in phagocytic cells [23]. Although the PM interaction with the skin is not completely understood, oxidative stress has been considered one of the main mechanisms of action of particulate matter leading to skin toxicity [24,25,26]. Importantly, it is widely recognized that inflammation and oxidative stress play a pivotal role in the induction and progression of numerous skin conditions including premature skin aging, psoriasis, atopic dermatitis, and skin cancer [27,28,29,30].

In this study, we examined the impact of UVA-visible light on the toxicity of fine particulate matter (PM_2.5_) using human epidermal keratinocyte cell line (HaCaT) as a model of human epidermis. Results of our study demonstrated that irradiation of the cells containing PM_2.5_, with UVA-visible light significantly decreased the cell viability. EPR spin-trapping and time-resolved near-infrared phosphorescence measurements revealed that irradiated ambient particles generated free radicals and singlet oxygen which could be involved in PM-dependent phototoxicity. These reactive oxygen species may lead to oxidative damage of key cellular constituents including cell organelles and increase the activity of pro-apoptotic and pro-inflammatory markers.

## 2. Results

### 2.1. Size Analysis of PM Particles

Figure 1 shows filters containing PM_2.5_ particles collected in different seasons before isolation (Figure 1A), followed by a histogram of the particle size distribution (Figure 1B). As evident, all particles exhibited a heterogeneous size with multiple peaks being visible. In the case of the winter sample, peak maxima were at 23 nm, 55 nm, and 242 nm. For the spring sample, peak maxima were at 49 nm and 421 nm. For the summer sample, peak maxima were at 35 nm, 79 nm, 146 nm and 233 nm. For the autumn sample, peak maxima were at 31 nm, 83 nm, and 533 nm. Overall, particles from winter had the smallest size, whereas particles from spring had the largest size with particles from autumn and summer being in between. However, it should be noted that DLS cannot be used for the precise determination of the size of polydisperse samples, such as PM particles. Therefore, for a more precise size analysis we employed AFM imaging. Figure 1 shows representative topography images of PM_2.5_ particles isolated from different seasons (Figure 1C). It is apparent that the winter sample contained the smallest particles and was most homogeneous, whereas both spring and summer particles contained the largest particles and were very heterogeneous. The autumn sample on the other hand contained particles larger than the winter sample, but smaller than both spring and summer and was also much more homogenous than the latter samples.

### 2.2. Phototoxic Effect of Particulate Matter

To determine the phototoxic potential of PM two independent tests were employed: PI staining (Figure 2A) and MTT assay (Figure 2B). PM from all seasons, even at the highest concentrations used, did not show any significant dark cytotoxicity (Figure 2A). After irradiation, the viability of the cells was reduced in cells incubated with winter, summer, and autumn particles. In the case of summer and autumn particles, a statistically significant decrease in the cell survival was observed for PM concentration: 50 μg/mL and 100 μg/mL Irradiated cells, containing ambient particles collected in the winter showed reduced viability for all particle concentrations used, and with the highest concentration of the particles the cell survival was reduced to 91% of control cells. Due to the obvious limitation of the PI test, which can only detect necrotic cells, with severely disrupted membranes, the MTT assay, based on the metabolic activity of cells, was also employed (Figure 2B). Ambient particles inhibited metabolism not only of the irradiated cells but also in the control non-irradiated cells. However, the inhibitory effect was significantly more pronounced in irradiated cells. The most pronounced effect was observed in cells incubated with 100 μg/mL of winter particles, where the viability was reduced by 40% after 2-h irradiation, followed by summer and autumn particles which decreased the viability by about 30%.

### 2.3. Photogeneration of Free Radicals by PM

Many compounds commonly found in ambient particles are known to be photochemically active, therefore we have examined the ability of PM_2.5_ to generate radicals after photoexcitation at different wavelengths using EPR spin-trapping. The observed spin adducts were generated with different efficiency, depending on the season the particles were collected, and the wavelength of light used to excite the samples. (Appendix A). Importantly, no radicals were trapped where the measurements were conducted in the dark. 

All examined PM samples photogenerated, with different efficiency, superoxide anion. This is concluded based on simulation of the experimental spectra, which showed a major component typical for the DMPO-OOH spin adduct: (A_N_ = 1.327 ± 0.008 mT; A_H__α_ = 1.058 ± 0.006 mT; A_H__β_ = 0.131 ± 0.004 mT) [31,32]. The photoexcited winter and autumn samples also showed a spin adduct, formed by an interaction of DMPO with an unidentified nitrogen-centered radical (Figure 3A,D,E,H,I,L). This spin adduct has the following hyperfine splittings: (A_N_ = 1.428 ± 0.007 mT; A_H__α_ = 1.256 ± 0.013 mT) [31,33]. The autumn PMs, after photoexcitation, exhibited spin adducts similar to those of the winter PMs. Both samples, on top of the superoxide spin adduct and nitrogen-centered radical adduct, also showed a small contribution from an unidentified spin adduct (A_N_ = 1.708 ± 0.01 mT; A_H__α_ = 1.324 ± 0.021 mT). Spring (Figure 3B,F,J) as well as summer (Figure 3C,G,K) samples photoproduced superoxide anion (A_N_ = 1.334 ± 0.005 mT; A_H__α_ = 1.065 ± 0.004 mT; A_H__β_ = 0.137 ± 0.004 mT) and an unidentified sulfur-centered radical (A_N_ = 1.513 ± 0.004 mT; A_H__α_ = 1.701 ± 0.004 mT) [31,34]. Furthermore, another radical, probably carbon-centered, was photoinduced in the spring sample (A_N_ = 1.32 ± 0.016 mT, A_H__α_ = 1.501 ± 0.013 mT). The intensity rates of photogenerated radicals decreased with longer wavelength reaching very low levels at 540 nm irradiation making it impossible to accurately identify (Appendix A).

The kinetics of the formation of the DMPO adducts is shown in Figure 4. The first scan for every sample was performed in the dark and then the appropriate light diode was turned on. As indicated by the initial rates of the spin adduct accumulation, superoxide anion was most efficiently produced by the winter and summer samples photoexcited with 365 nm light and 400 nm (Figure 4A,C,E,G). Interestingly, while the spin adduct of the sulfur radical formed in spring samples, photoexcited with 365 and 400 nm, after reaching a maximum decayed with further sample irradiation (Figure 4B,F), in the summer sample, the same spin adduct exhibited monophasic kinetics (Figure 4C,G). The signal of *N*-centered radical was constantly growing during the irradiation and was significantly higher for the winter PM_2.5_ (Figure 4A) compared to autumn PM_2.5_ (Figure 4B) excited with 365 nm light and reaching similar values for 400 nm (Figure 4E,H) and 440 nm (Figure 4I,L) excitation. The unidentified radical (A_N_ = 1.708 ± 0.01 mT; A_H__α_ = 1.324 ± 0.021 mT) produced by photoexcited winter and autumn particles demonstrated a stable growth for examined samples, with a biphasic character for winter PM_2.5_ irradiated with 365 nm (Figure 4A) and 400 nm (Figure 4E) light. Another unidentified radical, produced by spring PM_2.5_, that we suspect to be carbon-based (A_N_ = 1.32 ± 0.016 mT, A_H__α_ = 1.501 ± 0.013 mT), exhibited a steady increase during the irradiation for all examined wavelengths (Figure 4B,F,J). The initial rates of the radical photoproduction were calculated from exponential decay fit and were found to decrease with the wavelength-dependent manner (Appendix A).

### 2.4. Photogeneration of Singlet Oxygen (^1^O_2_) by PM

To examine the ability of PM from different seasons to photogenerate singlet oxygen we determined action spectra for photogeneration of this ROS. Figure 5 shows absorption spectra of different PM (Figure 5A) and their corresponding action spectra for photogeneration of singlet oxygen in the range of 300–580 nm (Figure 5B). Perhaps not surprisingly, the examined PM generated singlet oxygen most efficiently at 300 nm. For all PMs, the efficiency of singlet oxygen generation substantially decreased at longer wavelengths; however, a local maximum could clearly be seen at 360 nm. The observed local maximum might be associated with the presence of benzo[a]pyrene or another PAH, which absorb light in near UVA [35] and are known for the ability to photogenerate singlet oxygen [10,11]. Although in near UVA, the efficiency of different PMs to photogenerate singlet oxygen might correspond to their absorption, no clear correlation is evident. Thus, while at 360 nm, the effective absorbances of the examined particles are in the range 0.09–0.31, their relative efficiencies to photogenerate singlet oxygen vary by a factor of 12. It suggests that different constituents of the particles are responsible for their optical absorption and photochemical reactivity. To confirm the singlet oxygen origin of the observed phosphorescence, sodium azide was used to shorten the phosphorescence lifetime. As expected, this physical quencher of singlet oxygen reduced its lifetime in a consistent way (Figure 5C).

### 2.5. Light-Induced Lipid Peroxidation by PM

In both liposomes and HaCaT cells, the examined particles increased the observed levels of lipid hydroperoxides (LOOH), which were further elevated by light (Figure 6). In the case of liposomes (Figure 6A), the photooxidizing effect was highest for autumn particles, where the level of LOOH after 3 h irradiation was 11.2-fold higher than for irradiated control samples without particles, followed by spring, winter and summer particles, where the levels were respectively 9.4-, 8.5- and 7.3-fold higher than for irradiated controls. In cells, the photooxidizing effect of the particles was also most pronounced for autumn particles, showing a 9-fold higher level of LOOH after 3 h irradiation compared with irradiated control. The observed photooxidation of unsaturated lipids was weaker for winter, spring, and summer samples resulting in a 5.6, 3.6- and 2.8-fold increase of LOOH, compared to control, respectively. Changes in the levels of LOOH observed for control samples were statistically insignificant. The two analyzed systems demonstrated both season- and light-dependent lipid peroxidation. Some differences in the data found for the two systems might be attributed to different penetration of ambient particles. Moreover, in the HaCaT model, photogenerated reactive species might interact with multiple targets besides lipids, e.g., proteins resulting in relatively lower LOOH levels compared to liposomes.

### 2.6. The Relationship between Photoactivated PM and Apoptosis

The phototoxic effect of PM demonstrated in HaCaT cells raised the question about the mechanism of cell death. To examine the issue, flow cytometry with Annexin V/Propidium Iodide was employed to determine whether the dead cells were apoptotic or necrotic (Figure 7A,B). The strongest effect was found for cells exposed to winter and autumn particles, where the percentage of early apoptotic cells reached 60.6% and 22.1%, respectively. The rate of necrotic cells did not exceed 3.4 % and did not vary significantly between irradiated and non-irradiated cells. We then analyzed the apoptotic pathway by measuring the activity of caspase 3/7 (Figure 7C). While cells kept in the dark exhibited similar activity of caspase 3/7, regardless of the particle presence, cells exposed to light for 2 h, showed elevated activity of caspase 3/7. The highest activity of caspase 3/7 (30% higher than in non-irradiated cells), was detected in cells treated with ambient particles collected in the autumn. Cells with particles collected in the summer, winter, and spring showed a 25%, 18%, and 7% increase of caspase 3/7 activity, respectively.

To get a better understanding of the apoptosis induced in the cells by the concerted action of light and ambient particles, levels of selected pro-apoptotic markers such as Caspase-9, Bax, and cell stress NF-κB were investigated using quantitative real-time PCR (Figure 8). It is apparent that the expression of Bax and Caspase-9 genes in cells containing the particles was elevated by light. The expression of Bax in non-irradiated cells did not differ significantly from the control. However, two-hour irradiation resulted in a significant increase in the expression of Bax in cells containing particles, with winter particles having the highest effect (Figure 8A). The expression of Caspase-9 was significantly elevated by light in cells containing particles collected in the winter, summer, and spring, with a rather modest increase observed for autumn particles (Figure 8B). NF-κB is a well-known protein complex which controls the transcription of DNA; the level of its expression increases in response to cell stress, cytokines, free radicals, heavy metals, and ultraviolet radiation [36]. Interaction of ambient particles with HaCaT cells leads to the activation of NF-κB in a dose-dependent manner (Figure 8C). However, the combined action of the particles and light irradiation had a much stronger effect on activation of NF-κB. The highest expression of this nuclear factor was found in irradiated cells exposed to winter ambient particles, followed by summer, autumn, and spring particulate matter.

Mitochondria play a crucial role in apoptosis induced by many stress factors. The data obtained by the MTT assay (Figure 2B) and the detected changes in the expression of apoptosis-related genes associated with mitochondrial stress (Figure 8A,B) justified measurements to determine if the examined particles induce changes in the mitochondrial membrane potential (MMP) using the JC-10 fluorescent probe (Figure 9). A decrease in the red/green fluorescence ratio, arising from a reduction of MMP, was observed in cells supplemented with the particles and irradiated with light. A 22% decrease in the JC-10 aggregate/JC-10 monomer ratio was found in HaCaT cells incubated with 100 μg/mL of winter ambient particles. A significant decrease in the fluorescence ratio was also observed for spring (14%) and autumn (11%) ambient particles. The smallest effect was found for particles obtained in the summer.

## 3. Discussion

According to the WHO report, 4.2 million deaths each year can be associated with ambient air pollution [3]. Moreover, the report also indicates that only 10% of the world’s population lives in cities that comply with the recommended air quality guidelines. In recent years, significant efforts were made to examine the biological consequences of exposure to ambient particulate matter. It was demonstrated that ambient particles might contribute to a range of diseases including cardiovascular disease, chronic bronchitis, diabetes, and cancer [37,38]. The recently investigated exposure of the skin to particulate matter led to a conclusion that ambient particles could penetrate both disrupted and non-disrupted skin, causing adverse effects including skin barrier dysfunction and ROS-dependent skin aging [15,16,17]. In this study, we focused on the light-induced toxicity mediated by PM_2.5_ obtained in different seasons. 

The composition of ambient particles plays a critical role in their toxicity. Due to redox properties, transition metal ions, such as iron and copper, can generate ROS, including the most reactive hydroxyl radicals, via interaction with hydrogen peroxide and molecular oxygen [39,40,41]. The toxic effects of ROS may be intensified by non-redox active metals such as lead or aluminum [42,43] that are also found in PM [44]. Highly lipophilic polycyclic aromatic hydrocarbons, (PAHs), can efficiently penetrate the skin [45] and activate the aryl hydrocarbon receptor (AhR) in keratinocytes and melanocytes [46]. The activation of AhR was found to upregulate the expression of cytochrome P450 and promote intracellular oxidative stress [47]. Importantly, elevated cutaneous levels of reactive oxygen species were found to trigger a permanent pro-oxidative condition known as OxInflammation, which can lead to chronic systemic or local damage because of the crosstalk between oxidative stress and inflammatory mediators [48]. We are aware of only a single study that reported on the synergistic effect of pollutants and UV radiation on skin damage [49]. However, the cited study, which focused on the combined action of ozone and diesel engine exhaust (DEE) particles photoactivated by UVB/UVA radiation, is of limited relevance to the phototoxic potential of ambient particles under typical environmental conditions.

The formation of different radicals, induced by UV/visible light irradiation of ambient particles, might be attributed to their different sources responsible for different compositions of air pollution during different times of the year [50,51,52]. Although previous studies showed that particulate matter could generate superoxide anion, hydroxyl radicals, and carbon-centered radicals [53,54], we have demonstrated that PM_2.5_, upon irradiation with UV/visible light, can also generate nitrogen- and sulfur-centered radicals (Figure 3 and Figure 4). A high concentration of DMSO used in our EPR-spin trapping measurements excluded the possibility of detecting DMPO-OH, even if hydroxyl radicals were formed by photoexcitation of the ambient particles. It has previously been shown that the fast interaction of DMSO with OH leads to the formation of secondary products—methane sulfonic acid and methyl radicals [55,56]. It cannot be ruled out that the unidentified spin adduct observed during irradiation of winter, spring, and autumn particles was due to the interaction of DMPO with a carbon-centered radicals such as CH_3_. 

We have shown that both the levels and kinetics of free radicals photoproduction by PM_2.5_ are strongly season- and wavelength-dependent (Figure 4), with the highest values found for winter particles excited with 365 nm light. The highest phototoxicity and photoreactivity of the winter particles could be due to the fact that winter is the heating season in Krakow, during which burning coal generates a significant amount of air pollution [50,51,52]. Therefore, the winter particles are likely to contain a substantial amount of highly photoreactive aromatic hydrocarbons. The highest integrated absorption of winter particles in the UVA-blue part of the spectrum is consistent with such explanation. Another factor that could contribute to the higher photoreactivity of the winter particles is their smaller size and thus the higher surface to volume ratio when compared to the particles collected in other seasons.

Several chemicals commonly present in the particulate matter, particularly PAHs, are known to act as photosensitizing agents efficiently photogenerating singlet oxygen [6,7,9] by type II photooxidation. In a recent study, Mikrut et al. demonstrated that samples of ambient particles produced singlet oxygen upon irradiation with 290 nm light [54]. Although that observation indicated the photoreactivity of PM, it is of little biological relevance considering that no more than 5% of the UVB (280–315 nm) reaches the Earth’s surface [57]. Moreover, most of the UVB radiation is dissipated in the stratum corneum of the skin and practically no UVB penetrates viable parts of the epidermis [14,58]. Employing time-resolved singlet oxygen phosphorescence, we have proved that ambient particles can photogenerate singlet oxygen even when excited with 440 nm light (Figure 5). Singlet oxygen is viewed as one of the key reactive oxygen species responsible for cellular damage associated with so-called photodynamic action [59,60]. The highest phototoxicity found for winter PM_2.5_ coincided with their highest efficiency to photogenerate singlet oxygen, which could be partially explained by the smaller size of the particles and thus the highest surface to volume ratio, when compared to the particles collected in other seasons 

The demonstrated photogeneration of free radicals and singlet oxygen by short wavelength-visible light and, in particular, by long-wavelength UVA, is interesting and could be of importance considering the phototoxic potential of ambient particles and the ability of near UVA and blue light from solar radiation to penetrate human epidermis [14,58], as well as the increasing exposure of the human skin to short-wavelength visible light from artificial sources. These particles were also shown to effectively photogenerate superoxide anion as well as *N*-centered radical (Figure 3 and Figure 4 and Appendix A). Although it is tempting to speculate that these reactive oxygen species may determine the phototoxic potential of the studied ambient particles, a word of caution is necessary when comparing photobiological effects with photochemical phenomena. Thus, the highest efficiency to mediate photoperoxidation of unsaturated lipids was found for autumn particles (Figure 6). It is unknown why the higher flux of ROS photogenerated by winter particles, compared to other particles, did not induce the highest peroxidation of lipids in liposomes and in HaCaT cells. 

The relationship between the concentration of ambient particles and the viability of HaCaT cells was previously investigated. Thus, Li et al. demonstrated that the cytotoxicity of ambient particles against HaCaT cells was dose-dependent in a range of 0–800 μg/mL [61]. Romani et al. showed that time of the exposure to Concentrated Air Particles (CAPs) was a crucial factor for toxicity against HaCaT cells [26]. Another group demonstrated significant dark cytotoxicity of 100 µg/mL PM_2.5_ [62]; however, no cytotoxicity was observed at very low doses (up to 200 ppm) [63]. In this study, we have confirmed the dark toxicity of PM_2.5_, especially when high concentrations of the particles were used. We have also demonstrated the highest phototoxicity of the particles collected during the winter (Figure 2A,B). Photoactivation of the particles with UVA-vis light from the solar simulator markedly increased the toxicity of particles, as demonstrated by MTT assay (Figure 2B). Flow cytometry measurements performed immediately after irradiation confirmed that the exposure of HaCaT cells to PM_2.5_ in the dark and, particularly, after light treatment resulted predominantly in the apoptotic pathway of the cell death, with very little or no necrosis observed (Figure 7A,B). While the late stage of apoptosis is associated with disruption of the cell mitochondria in a necrotic pathway, cell membrane integrity is lost, facilitating the influx of propidium iodide into the cell and binding to DNA [64]. It must be stressed that flow cytometry experiments were performed directly after light treatment, whereas both viability assays were conducted 24 h after the treatment to examine both lethal and sub-lethal damage induced by photoexcited ambient particles. Therefore, it might be reasonable to assume that a greater part of early apoptotic cells detected using flow cytometry progressed during 24 h entering the late stage of apoptosis, where the functions are lost. Our results are in agreement with previous studies which indicated that exposure to PM in the dark can induce apoptosis in HaCaT cells in a dose-dependent manner [65,66]. Moreover, we have shown that light irradiation leads to a significant increase in the number of apoptotic cells compared to non-irradiated samples exposed to PM_2.5_.

It is apparent that cell exposure to particulate matter does not significantly increase the levels of Bax expression. However, light treatment resulted in a considerable rise in the gene expression of Bax (Figure 8A). Overexpression of Bax protein resulted in the condensation, fragmentation, and clustering of mitochondria and lost of their metabolic activity, which was found in an independent study [67]. It is in agreement with the results of the MTT assay presented in this study (Figure 2B), where the decreased metabolic activity causing increased cell mortality correlated with elevated levels of Bax. The interaction of particulate matter with UV-vis light was also found to cause a considerable increase of caspases 3/7, and 9 activity (Figure 7C and Figure 8B), consistent with the results discussed above. Specific components of particulate matter can trigger intracellular oxidative stress promoted by the activation of NF-kB signaling [47,68,69]. We have demonstrated that co-exposure of HaCaT cell to PM_2.5_ and light result in a significant increase of NF-kB gene level (Figure 8C). Therefore, we postulate that the demonstrated effect, when persisting for a longer time, might result in OxInflammation—a pro-oxidative feature leading to chronic pathological conditions [48]. 

Mitochondria were previously demonstrated to be a target of environmental pollutants including particulate matter [70]. Exposure of HaCaT cells to PM_2.5_ leads to the induction of oxidative stress [71,72] that promotes mitochondria swelling, resulting in deregulation of the mitochondrial respiratory chain and production of ROS [70]. In this study, we observed that cells incubated with PM_2.5_ and kept in the dark exhibited only a limited reduction in MMP. However, cells exposed to light from the solar simulator exhibited significantly lower MMP compared to non-irradiated cells (Figure 9). Since the disruption of mitochondria plays an important role in the induction and progression of various skin diseases [73], including skin cancer, the obtained data support the hypothesis of a possible involvement of light-induced PM_2.5_ in skin pathologies.

Lipids found in epidermal keratinocytes play a crucial role in forming the skin barrier against microorganisms, pollution, and maintaining homeostasis [74,75]. Due to their essential role, the effect of PM_2.5_ exposure on the properties of epidermal lipids was previously investigated [68,71,76]. Using the fluorescent probe DPPP and a specific lipid peroxides marker 8-isoprostane, PM_2.5_ was found to induce lipid peroxidation [71,76]. The in vivo lipid peroxidation was previously demonstrated in an HR-1 mouse (hairless male mice) model, where 100 µg/mL of PM_2.5_ was dispersed in propylene glycol, applied over 1 cm^2^ area of dorsal skin for 7 consecutive days and the exposed skin tissue was analyzed using DPPP probe [70]. In our study, we have employed liposomes as a simple model of cellular lipid membrane to demonstrate that the activation of PMs by light from solar simulator can significantly promote oxidation of unsaturated lipids (Figure 6A). The photoperoxidizing ability of the studied PMs was confirmed in HaCaT cells used as an in vitro model of the skin epidermis (Figure 6B). Based on the acquired data, we postulate that mitochondria and lipids may act as potential targets of phototoxicity mediated by PM in skin cells.

We have demonstrated that light interacting with particulate matter increases the damage of skin cells in vitro. For the first time, we present season-dependent and light-dependent effect of fine particulate matter on viability of HaCaT cells, apoptotic cell death, lipid peroxidation, and mitochondrial membrane potential. We hypothesize that photoproduction of free radicals and singlet oxygen is, in part, responsible for the observed biological response.

## 4. Materials and Methods

### 4.1. Materials

The following chemicals were obtained from Sigma-Aldrich (Steinheim, Germany): 3-(4,5-dimethylthiazol-2-yl)-2,5-diphenyltetrazolium bromide (MTT), Dulbecco’s Modified Eagle Medium (DMEM) with and without phenol red, propidium iodide (PI), Triton X-100, dichloromethane (DCM), hexane (Hx), L-α-phosphatidylcholine (L-α-PC) from chicken’s egg, chloroform, tert-Butyl hydroperoxide solution, cadmium acetate, and deuterium oxide. 5,5-Dimethyl-1-Pyrroline N-oxide (DMPO) was obtained from Dojindo (Kumamoto, Japan). Fetal bovine serum (FBS) was purchased from Gibco (Carlsbad, CA, USA). Potassium iodide was purchased from Chempur (Piekary Slaskie, Poland). Acetic acid and dimethyl sulfoxide (DMSO) were purchased from POCH (Gliwice, Poland). Alexa Fluor 488 Annexin V/Dead Cell Apoptosis Kit was purchased from Life Technologies (Carlsbad, CA, USA). Caspase-Glo^®^ 3/7 was purchased from Promega (Madison, WI, USA). JC-10 Mitochondrial Membrane Potential Assay Kit was purchased from Abcam (Cambridge, UK). RNA Extracol, NG dART RT kit, and SG qPCR Master Mix (2×) were obtained from EURx (Gdansk, Poland). 

### 4.2. Particulate Matter Extraction

Filters containing PM particles of a size below 2.5 µm collected in Cracow using low volume LVS-3 samplers with 2.3 m^3^/h flow rate (24 h exposure) were obtained from the Environmental Protection Inspectorate (WIOŚ) in Cracow. Filters were divided into 4 groups depending on the season of the year 2019: winter (December to February), spring (March to May), summer (June to August) and autumn (September to November). PM was extracted from filters based on a previously described method [77]. Extraction of PM procedure was carried out under low light condition.

### 4.3. Dynamic Light Scattering

Dynamic light scattering (DLS) was used to determine the size distribution of PM. Samples were diluted in distilled water to a final concentration of 0.1 mg/mL and analyzed using Zetasizer Nano S (Malvern Panalytical, Malvern, UK) as described previously [78,79].

### 4.4. Atomic Force Microscopy

Atomic force microscopy (AFM) was used to image particles obtained from different seasons. For the analysis, a small droplet of each sample was placed on freshly cleaved mica surface and evaporated in a desiccator. Topography images of the particles were obtained in PeakForce Tapping mode employing the BioScope Catalyst AFM from Bruker. ScanAsyt-Air probes with a nominal tip radius of 2 nm and a spring constant of 0.4 N/m were used (Bruker Probes). Details on AFM analysis can be found elsewhere [80].

### 4.5. Cell Treatment and Light Irradiation

Human epidermal keratinocytes (HaCaT cell line) were passaged weekly and kept in high glucose DMEM culture medium supplemented with 10% fetal bovine serum (FBS) and antibiotics (penicillin 150 U/mL, streptomycin 100 μg/mL) under 37 °C in a 5% CO_2_ humidified atmosphere. After reaching confluency, cells were seeded into 96 or 24 well plates and incubated with predetermined concentrations of PM in culture medium for 24 h. To examine the phototoxic effect of PM on the cells, the particles were used at the concentration: 25, 50, and 100 μg/mL. After 24 h of incubation with PM, cells were irradiated for 1 or 2 h using a SS1.6 kW solar simulator (ScienceTech, London, Ontario, Canada) set to 1250 W equipped with AirMass 0 filter (ScienceTech, London, Ontario, Canada) and 330 nm cut-off filter. Spectral irradiance of the light used in the experiments is shown in Appendix A. Shortly before irradiation, culture media were exchange with similar media deprived of phenol red and supplemented with 2% FBS. During irradiation, cells were placed on a cooling plate providing stable temperature. Immediately after irradiation, the culture media were changed for the initial media. Control, non-irradiated cells underwent similar media exchange as irradiated cells. 

### 4.6. Propidium Iodide Staining

Survival of the cells was confirmed 24 h after irradiation by quantifying nuclei in the cells using a membrane permeable fluorescent dye propidium iodide (PI) as described previously [81]. The number of PI-positive nuclei was quantified using a custom written script for ImageJ software (National Institutes of Health, Bethesda, MD, USA). The number of viable cells per field was expressed as a percent of the total cell number determined by adding Triton X-100 at a final concentration of 0.1% and kept for 10 min after which fluorescence images from the same area were recorded. The experiments were repeated three times. 

### 4.7. MTT Assay

The cytotoxic effect of light irradiation was determined 24 h after the irradiation using MTT assay as described previously [82]. In brief, MTT reagent diluted in DMEM culture medium was added to control and treated cells. After incubation for 20 min at 37 °C, culture medium was removed, and the remaining blue formazan crystals were solubilized in DMSO/ethanol (1:1). The absorbance was detected at 560 nm using a plate reader (GENios Plus, Tecan, Austria GMbH) and results were reported as a percent of untreated controls. The experiments were repeated three times for statistics.

### 4.8. Detection of Free Radicals by EPR Spin Trapping

EPR spin trapping was employed to detect light-induced radicals using 100 mM DMPO as a spin trap. Samples containing the spin trap and suspension of particulate matter (0.25 mg/mL) in 70% DMSO/30% H_2_O [83] were irradiated in EPR flat cell in the resonant cavity with UVA (365 nm, 10 mW/cm^2^), violet-blue light (400 nm, 10 mW/cm^2^), blue light (440 nm, 10 mW/cm^2^) or green light (540 nm, 10 mW/cm^2^) using dedicated custom-made high-power LED chips (CHANZON, China) with home built cooling systems. The EPR measurements were carried out employing a Bruker-EMX AA spectrometer (Bruker BioSpin, Germany), using the following apparatus settings: 10.6 mW microwave power, 0.05 mT modulation amplitude, 332.4 mT center field, 8 mT scan field, and 84 s scan time. Simulations of EPR spectra were performed with EasySpin toolbox for MATLAB [84]. The EPR spin trapping measurements were repeated 3 times.

### 4.9. Time-Resolved Detection of Singlet Oxygen Phosphorescence

D2O suspension of PM (0.2 mg/mL) in a 10-mm optical path quartz fluorescence cuvette (QA-1000; Hellma, Mullheim, Germany) was excited for 30 s with laser pulses generated by an integrated nanosecond DSS Nd:YAG laser system equipped with a narrow-bandwidth optical parameter oscillator (NT242-1k-SH/SFG; Ekspla, Vilnius, Lithuania), operating at 1 kHz repetition rate. The near-infrared luminescence was measured perpendicularly to the excitation beam using a thermoelectric cooled NIR PMT module (H10330-45; Hamamatsu, Japan) equipped with a 1100-nm cut-off filter and dichroic 1270 nm filter. Signals were collected using a computer-mounted PCI-board multichannel scaler (NanoHarp 250; PicoQuant GmbH, Berlin, Germany) [85]. During measurements, the samples were constantly stirred using a miniature magnetic stirrer. The singlet oxygen phosphorescence measurements were repeated three times for statistics.

### 4.10. Liposome Preparation and Iodometric Assay for Lipid Hydroperoxide Measurements

An iodometric assay was used to assess lipid peroxidation induced by light-excited PM. The assay was performed on cells and in model system. In the case of the former, HaCaT cells were incubated with solutions of PM in high glucose DMEM at a concentration of 100 µg/mL for 24 h, then growing medium was removed and the cells were collected in PBS using cell scraper. In a model system, lipids (l-α-phosphatidylcholine (PC) from chicken’s egg) were dissolved in chloroform, vortexed, evaporated under argon for 10–15 min and finally dried using a vacuum pump to form a lipid film. Next, suspension of PM in PBS at a concentration of 100 µg/mL were added to the lipids, frozen in liquid nitrogen and thawed at 40 °C to obtain liposomes with incorporated PM. For both liposomes and HaCaT cells, lipids were isolated after irradiation using Folch extraction procedure and chloroform phase was dried under stream of argon. To quantify lipid peroxides, samples were gently degassed with argon and suspended in acetic acid/chloroform solution (3:2). The potassium iodide solution (1.2 g/mL) was then added, gently mixed, and left for 10 min. After this time, 0.5% cadmium acetate in 0.1 M acetic acid was added to the solution. Tert-butyl hydroperoxide solutions were used to prepare calibration curve. To prevent oxidation of iodide ions by atmospheric oxygen, all used solutions were kept under argon. Finally, absorbance was measured at 352 nm against water sample using HP 8452 A spectrophotometer (Hewlett-Packard, Palo Alto, CA, USA). The iodometric assays were repeated three times for statistics.

### 4.11. Flow Cytometry

To quantify apoptotic and necrotic cells, flow cytometry was performed. HaCaT cells (1 × 10^6^ cells/sample) were washed twice with cold PBS immediately after irradiation and centrifuged at 1000× *g* for 5 min. Pellets were suspended in annexin binding buffer and cells were incubated with FITC annexin V and PI for 15 min in room temperature. Next, 10^4^ unfixed cells per sample was analyzed with flow cytometry (LSR Fortressa, BD, San Jose, CA, USA) as described in detail elsewhere [86]. Three independent experiments were performed.

### 4.12. Caspase 3/7 Fluorometric Analysis

Cell apoptosis was analyzed by the measurement of caspase 3/7 activity as described previously [86]. In brief, HaCaT cells (5 × 10^5^ cells/well) were placed in 96-well white-bottom microplate. Directly after irradiation, cells were washed with PBS and 100 µL of Caspase-Glo 3/7 reagent was added to each well. Finally, the plate was gently mixed by shaking at 200 rpm for 30 s and the chemiluminescence was measured continuously for 40 min at 37 °C. The assay was repeated three times.

### 4.13. Real-Time PCR

Immediately after the experiments, cells were washed twice with cold PBS and harvested in Extracol. The concentrations of isolated RNA were determined using NanoDrop™ One (DeNovix, Wilmington, DE, USA). 1 µg of RNA was reverse transcribed using NG dART kit in thermal cycling condition: 65 °C for 60 min, 85 °C for 5 min, and finally cooling to 4 °C. The RT-PCR was performed using 20 ng of cDNA, specific primers and SG qPCR Master Mix in Applied BiosystemsTM StepOne PlusTM Real-Time PCR (Applied Biosystems, Foster City, CA, USA) with thermal condition: 1 cycle of 95 °C for 15 min, 40 cycles of 95 °C for 60 s, 40 cycles of 60 °C for 30 s, and 1 cycle of 72 °C for 30 s. Values of mRNA expression for Bax, Caspase-9, and NF-kB were normalized to gene reference-elongation factor-2 (EF2). The relative gene expression was quantified using ΔΔCT method as described previously [81]. RT-PCR measurements were conducted 3 times for statistical purposes

### 4.14. Mitochondrial Membrane Potential Detection

The depolarization of mitochondrial membrane was assessed using the JC-10 assay. Control and treated cells were incubated with JC-10 dye-loading solution (50 µL/well/96-well plate) in DMEM culture medium without phenol red for 1 h at 37 °C, 5% CO_2_. Next, cells were washed in PBS and changes in fluorescence emission intensity were detected by a plate reader (ClarioStar, BMG Labtech, Cary, NC, USA) using the following settings: red-excitation/emission 560/595 nm; green-excitation/emission 485/535 nm. The mitochondrial membrane potential was presented as red/green ratio, where the decrease reflects mitochondrial depolarization. JC-10 assay was repeated 3 times.

### 4.15. Statistical Analysis

Each of the experiments were repeated at least three times, resulting in consistent results. Statistical analysis of the data was performed using OriginPro software (OriginLab, Northampton, MA, USA). Statistical significance was assessed by ANOVA with Tukey post-hoc test, and *p* values below 0.05 were considered as statistically significant. 

## 5. Conclusions

Our study has demonstrated that sunlight can significantly increase PM-mediated toxicity in skin cells. PM_2.5_ photogenerated free radicals and singlet oxygen in a season-dependent and wavelength-dependent manner. Photoexcited particles can cause skin damage through induction of oxidative stress, which promotes apoptotic cell death, decreases mitochondrial membrane potential, and induces peroxidation of intracellular lipids in a season-dependent way. Here, we showed, for the first time, the importance of the interaction of ambient particles and solar radiation for inducing potential damage to human skin.

## Figures and Tables

**Figure 1 ijms-22-10645-f001:**
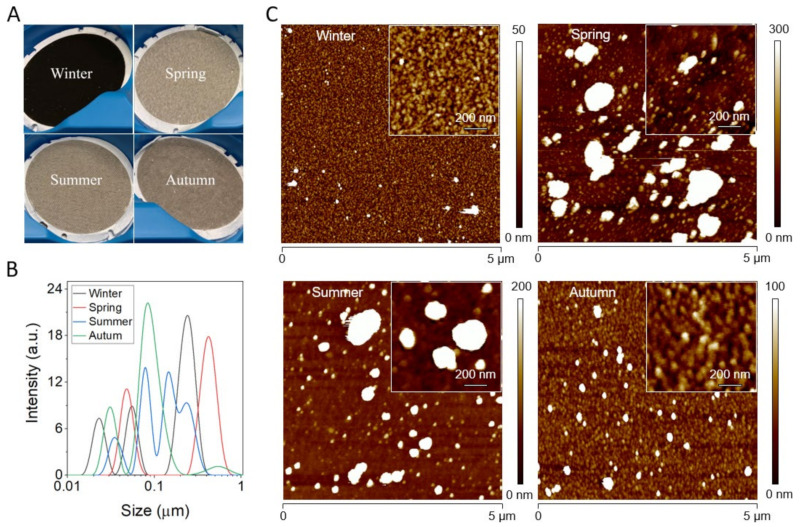
Characterization of PM particles. (**A**) Photos of filters containing PM_2.5_ particles before isolation. (**B**) DLS analysis of isolated particles: winter (black line), spring (red line), summer (blue line), autumn (green line). (**C**) AFM topography images of PM particles isolated from winter, spring, summer, and autumn samples. Insets show high magnification images of the particles.

**Figure 2 ijms-22-10645-f002:**
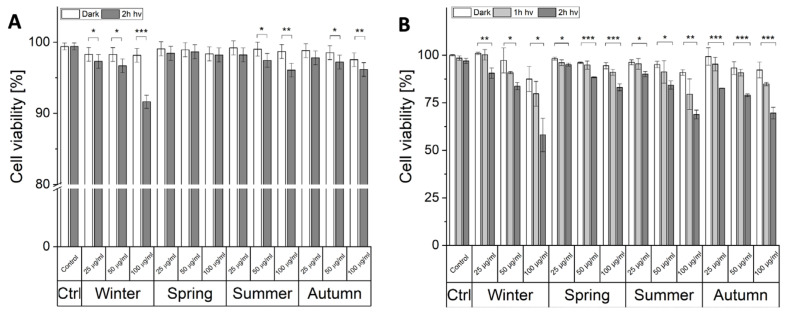
The photocytotoxicity of ambient particles. Light-induced cytotoxicity of PM_2.5_ using PI staining (**A**) and MTT assay (**B**). Data for MTT assay presented as the percentage of control, non-irradiated HaCaT cells, expressed as means and corresponding SD. Asterisks indicate significant differences obtained using ANOVA with post-hoc Tukey test (* *p* < 0.05, ** *p* < 0.01, *** *p* < 0.001). The viability assays were repeated three times for statistics.

**Figure 3 ijms-22-10645-f003:**
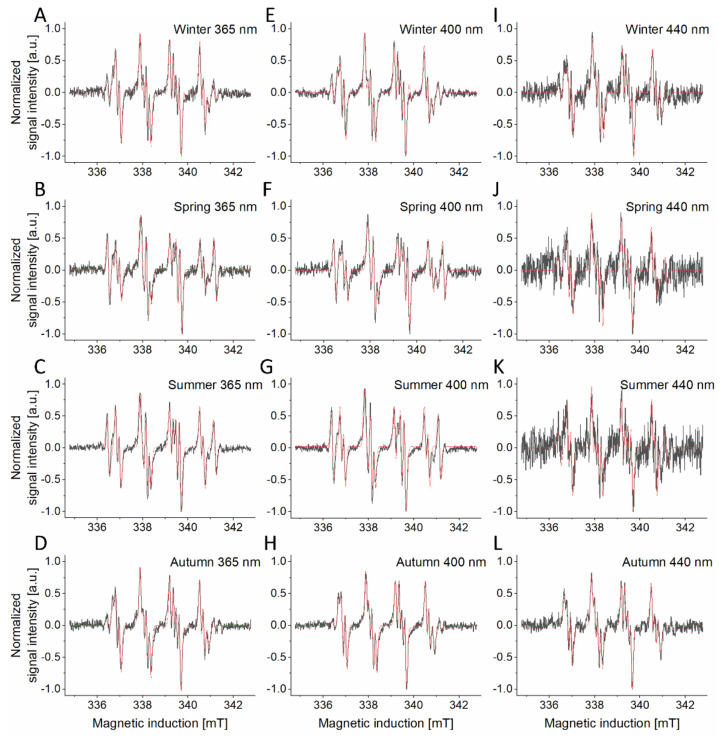
EPR spin-trapping of free radicals generated by PM samples from different seasons: winter (**A**,**E**,**I**), spring (**B**,**F**,**J**), summer (**C**,**G**,**K**) and autumn (**D**,**H**,**L**). Black lines represent spectra of photogenerated free radicals trapped with DMPO, red lines represent the fit obtained for the corresponding spectra. Spin-trapping experiments were repeated 3-fold yielding with similar results.

**Figure 4 ijms-22-10645-f004:**
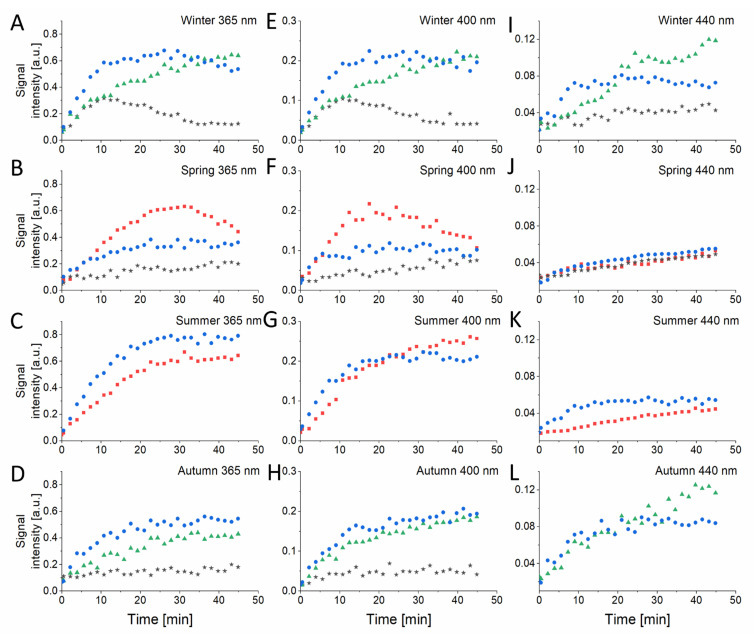
Kinetics of free radical photoproduction by PM samples from different seasons: winter (**A**,**E**,**I**), spring (**B**,**F**,**J**), summer (**C**,**G**,**K**) and autumn (**D**,**H**,**L**) obtained from EPR spin-trapping experiments with DMPO as spin trap. The radicals are presented as follows: superoxide anion–blue circles, *S*-centered radical–red squares, *N*-centered radical–green triangles, unidentified radicals–black stars.

**Figure 5 ijms-22-10645-f005:**
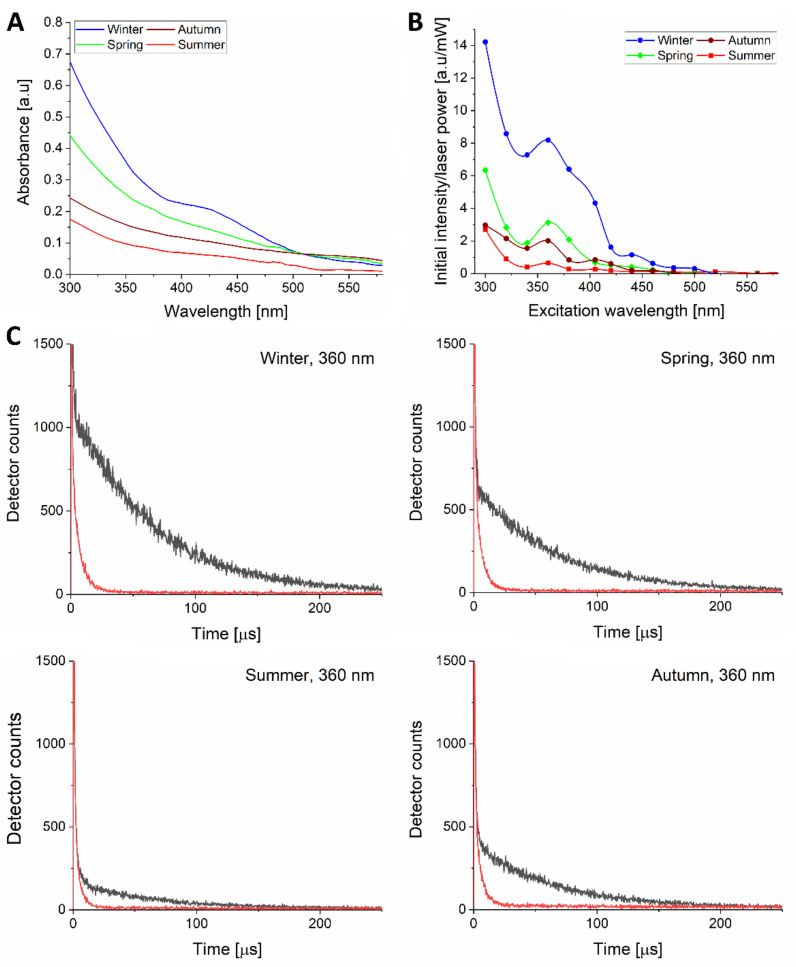
UV-Vis absorption spectra (**A**) and action spectra of singlet oxygen photogeneration (**B**) by 0.2 mg/mL of ambient particles: winter (blue circles), spring (green diamonds), summer (red squares), autumn (brown hexagons). Data points are connected with a B-spline for eye guidance. (**C**) The effect of sodium azide (red lines) on singlet oxygen phosphorescence signals induced by excitation with 360 nm light (black lines). The experiments were repeated 3 times yielding similar results and representative spectra are demonstrated.

**Figure 6 ijms-22-10645-f006:**
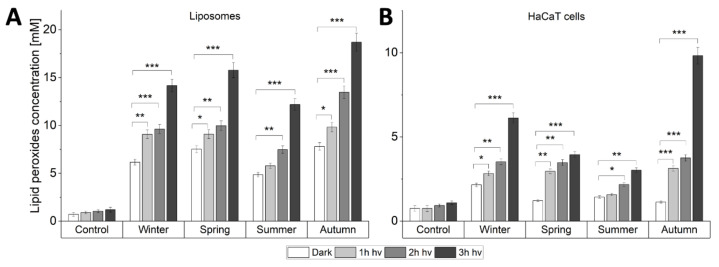
Lipid peroxidation induced by light-excited particulate matter (100 µg/mL) in (**A**) Liposomes and (**B**) HaCaT cells. Data are presented as means and corresponding SD. Asterisks indicate significant differences obtained using ANOVA with post-hoc Tukey test (* *p* < 0.05 ** *p* < 0.01 *** *p* < 0.001). The iodometric assays were repeated three times for statistics.

**Figure 7 ijms-22-10645-f007:**
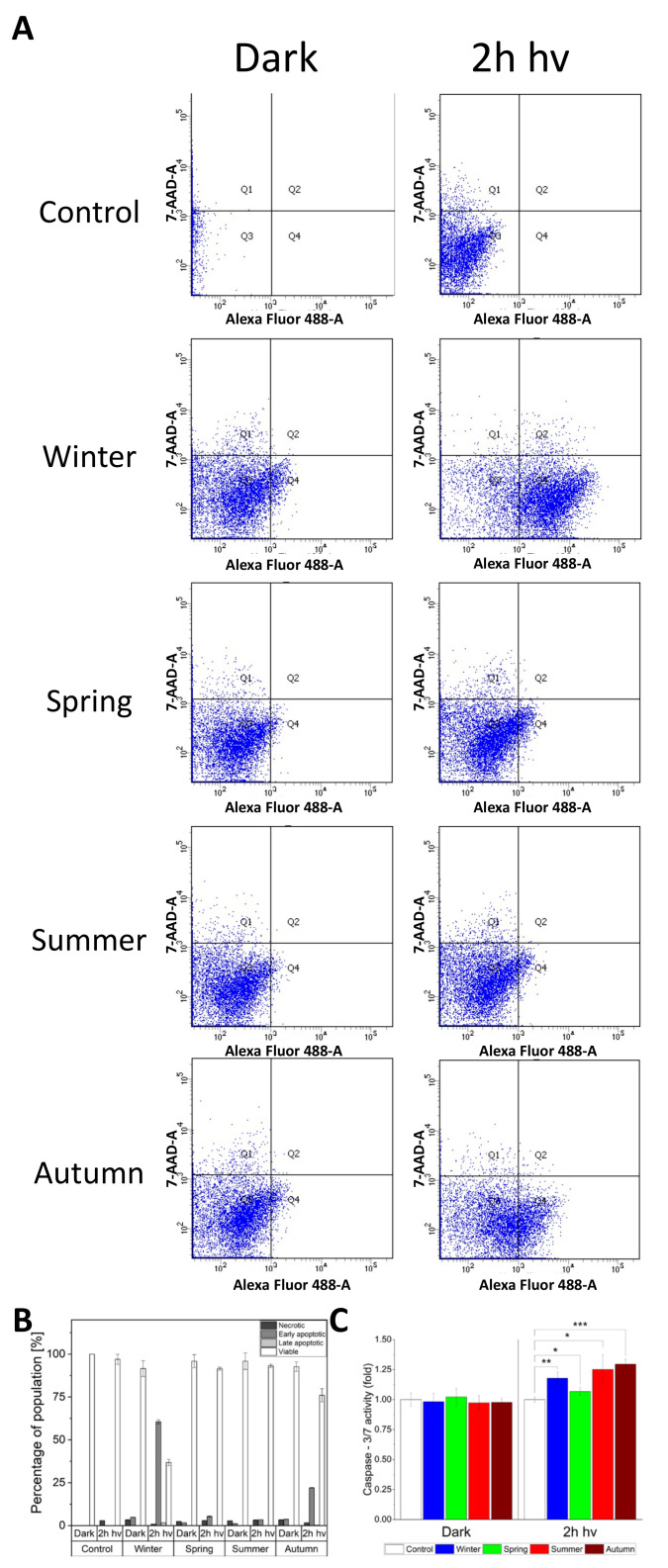
Examination of the cell death mechanism induced by light-irradiated PM from different seasons (100 μg/mL). (**A**) Flow cytometry diagrams representing Annexin V (AnV) and propidium iodide (PI) cell distribution. (**B**) The percentage ratio of signal detected for total cell population and showing no cell death (white bars), early apoptosis (dark grey bars), late apoptosis (light grey bars) and necrosis (black bars). For each sample, data were collected for 10^4^ HaCaT cells. (**C**) Caspase 3/7 activity in irradiated and non-irradiated cells incubated with ambient particles. All cells were incubated with Caspase-Glo-3/7 and chemiluminescence of samples was measured. Data are presented as means ± SD. Asterisks indicate significant differences obtained using ANOVA with post-hoc Tukey test (* *p* < 0.05, ** *p* < 0.01, *** *p* < 0.001). Flow cytometry experiments and Capase 3/7-assay were repeated three times.

**Figure 8 ijms-22-10645-f008:**
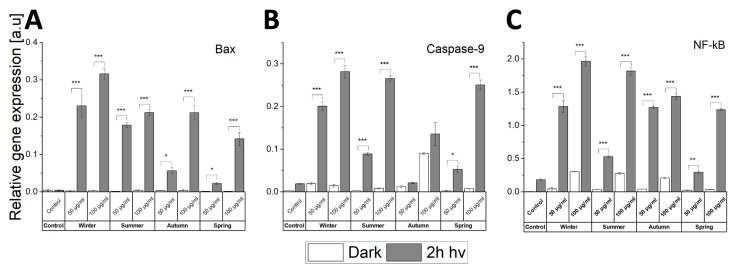
Relative gene expression of Bax (**A**), Caspase-9 (**B**), and NF-κB (**C**) determined using real-time PCR. HaCaT cells were exposed to PM_2.5_ (50 or 100 µg/mL) prior to 2 h light irradiation. Cells without ambient particles were used as controls. Data are presented as means ± SD. Asterisks indicate significant differences obtained using ANOVA with post-hoc Tukey test (* *p* < 0.05, ** *p* < 0.01, *** *p* < 0.001). RT-PCR experiments were conducted three times for statistics.

**Figure 9 ijms-22-10645-f009:**
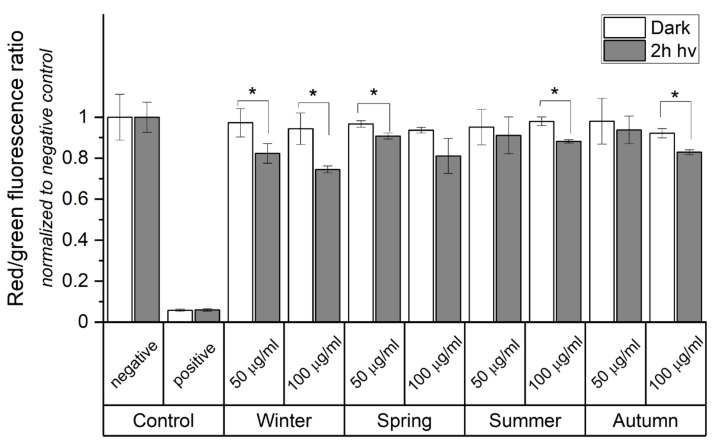
Change in mitochondrial membrane potential (MMP) determined by JC-10 assay. The diagram shows the quantitative ratio of JC-10 aggregates (red fluorescence) to JC-10 monomers (green fluorescence). Cells were exposed to PM_2.5_ (50 or 100 µg/mL) prior to 2 h light irradiation. Cells without ambient particles were used as negative controls. Cells incubated with 2% Triton X-100 were used as positive control. Data are normalized to dark, negative control and expressed as means ± SD. Asterisks indicate significant differences obtained using ANOVA with post-hoc Tukey test (* *p* < 0.05). JC-10 assay was repeated 3 times for statistics.

## Data Availability

The data presented in this study are available on request.

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
