# Peer review of "Fine Particulate Matter-Induced Oxidative Stress Mediated by UVA-Visible Light Leads to Keratinocyte Damage"

_ijms, 2021, doi:10.3390/ijms221910645_

Round 1

Reviewer 1 Report

The authors in the article ‘Light induced oxidative stress mediated by fine particle matter leads to keratinocyte damage’ studied a very interesting topic of the impact of air pollution on human health. The effect of particulate matter on skin health is really an important issue to investigate, and authors used a large panel of techniques to substantiate their conclusions. Even though in general I am in favour of this work, the presentation and, to a smaller extent also the interpretation of data requires some more work. Below I summarise all my concerns, which are related more to the quality of the manuscript than to the actual experiments. Therefore, I believe that the work is suitable for publication in IJMS journal after a rather intensive revision of the manuscript.

Major comments:

  1. If I understand it correctly PM2.5 fraction contains particles < 2.5 um and > 1 um. However, Fig. 1 shows almost exclusively particles with a particulate size < 1 um – in many critical cases reported in this work, the largest population has size < 0.1 um. Shouldn’t it be specified as a PM1 fraction?
  2. Could authors specify why PI (propidiumiodide) cannot label apoptotic cells? I have different experience.
  3. It is important to explain for a reader why samples were exposed to 1,2 4 … hours of “sunlight’ with a nominal energy 1.6 kW.
  4. In the Introduction authors discuss penetration of blue (~470 nm) and green light (> 500 nm) into the skin tissue. However, the most of the photoinduced changes were caused by the light of lower wavelengths. What is skin penetration efficiency of the light with wavelengths of 360, 400 or 440 nm? Are such data available? Please, discuss more in the Discussion section.
  5. The same Y-axis scales should be used for all graphs in Fig. 4.
  6. Sodium azide experiment data should be presented in Fig. 5 (as section c ?).
  7. Section 2.6 should go after section 2.2 - viability and cell death mechanism. Please, consider the change.
  8. Lines 249-251: Expression of changes in fold increase is an over-exaggeration. Obviously, if a starting point has very low value, the increase in signal/change is dramatic in terms of FOLD-increase. For cell death experiments (Fig. 7a,b), one should always report about the proportion of cells.
  9. 7: Dot plots: axis labels are completely invisible.
  10. CRITICAL: Fig. 8 (RT-PCR data): Do you have any control, which was not affected by the PM+light? Please, specify.
  11. Please, add some concise conclusions at the end of the discussion section.
  12. Section 4.9 – method: Please add all the details to the procedure. In the age of digital publishing, there is no need to refer to the previous work without mentioning all the details.
  13. Section 4.10: Why liposomes were not created first and then treated with PM? Post prep addition is more physiologically relevant.
  14. Introduction: The reference 14 reports on Monte Carlo simulations. It would be better to mention this in the text. Currently, the sentence (lines 58-60) sounds as this was already experimentally proven.

Mino comments/typos:

  1. The last sentence of the Abstract: please, avoid the word ‘clearly’. It is not yet that clear.
  2. Line 129: Which cells?
  3. Line 136: Photochemically
  4. Line 218: … with a B-spline for eye guidance.
  5. Line 219: … was repeated 3 times …
  6. Line 223: LOOH – explain the abbrev.
  7. Line 258: … activity of caspase …
  8. 7, 8, 9: How many independent experiments. Please, add to the figure captions.
  9. Line 275: … using quantitative real-time PCR …
  10. Line 312: … ratio of JC-10 …
  11. LINE 444 POLLUTANTS

Reviewer 2 Report

This study aimed to investigate the effects of light on the toxicity of fine particulate matter (PM2.5) in cultured human keratinocytes. The scope of this work is interesting. However, there are some comments as following:

  1. Since the authors reported the modulatory effects of UVA-visible light on PM-induced toxicity, the title should be “Fine particulate matter-induced oxidative stress mediated by visible light leads to keratinocyte damage”.
  2. Did the authors analyze the chemical composition of PM2.5 samples in different seasons? These data will be useful to explain the varying ability of PM to photogenerate ROS.
  3. Do the authors have any suggestion for the highest lipid peroxidation in autumn samples but not winter samples?
  4. Fig. 8 presents apoptosis-related genes rather than mitochondria (Line 300).

5. The authors should check the English throughout the manuscript, e.g. “pollutants” not “pullutanants” (Line 444)
